# Highly tunable band structure in ferroelectric R-stacked bilayer WSe$_2$

Zhe Li [1], Prokhor Thor[1], George Kourmoulakis[1], Tatyana V. Ivanova[1], Takashi Taniguchi [2], Kenji Watanabe [3], Hongyi Yu [4,5], Mauro Brotons-Gisbert [1] & Brian D. Gerardot [1] ✉

Transition metal dichalcogenide homobilayers unite two frontiers of quantum materials research: sliding ferroelectricity, arising from rhombohedral stacking, and moiré quantum matter, emerging from small-angle twisting. The spontaneous polarization of ferroelectric rhombohedral stacked homobilayers produces a highly tunable band structure, which, together with strain-induced piezoelectricity, governs the topology and correlated electronic phases of twisted bilayers. Here we present a systematic low-temperature optical spectroscopy study of rhombohedral stacked bilayer WSe$_2$ to quantitatively establish its fundamental electronic and ferroelectric properties. Exciton and exciton-polaron spectroscopy under doping reveals a pronounced electron-hole asymmetry that confirms type-II band alignment, with the conduction and valence band edges located at the Λ and K valleys, respectively. Through distinct excitonic responses and tunable interlayer-intralayer exciton hybridization under displacement fields, we uncover the coexistence of AB and BA ferroelectric domains. Using exciton-polarons as a probe, we directly measure the intrinsic polarization field and extract the interlayer potential. Finally, we demonstrate electric-field-driven symmetric switching of the valence band maximum, attributed to ferroelectric domain switching. These results provide a complete experimental picture of the band alignment, spontaneous polarization field, and domain dynamics of rhombohedral stacked WSe$_2$, establishing key parameters to understand twisted bilayers and enabling new ferroelectric and excitonic device opportunities.

Transition metal dichalcogenide (TMD) homobilayers exhibit a notable duality: rhombohedral (R) stacking generates intrinsic sliding ferroelectricity, while small twist angles give rise to moiré quantum phenomena. In R-stacked bilayers, two mirror-related layer registries - AB and BA (Fig. 1a) - break inversion symmetry and generate a spontaneous out-of-plane polarization[1]. The permanent dipole orientation

can be reversed by lateral sliding of one layer with respect to the other, giving rise to sliding ferroelectricity[2,3]. By exploiting both charge and layer degrees of freedom, this unique switching mechanism enables the design of new multiferroic devices for ultrafast, non-volatile memory, distinguishing it from conventional ferroelectrics, where atoms move along the field direction[4–6]. Recent experiments have

---

[1]Institute of Photonics and Quantum Sciences, SUPA, Heriot-Watt University, Edinburgh, UK. [2]Research Center for Materials Nanoarchitectonics, National Institute for Materials Science, 1-1 Namiki, Tsukuba, Japan. [3]Research Center for Electronic and Optical Materials, National Institute for Materials Science, 1-1 Namiki, Tsukuba, Japan. [4]Guangdong Provincial Key Laboratory of Quantum Metrology and Sensing & School of Physics and Astronomy, Sun Yat-Sen University (Zhuhai Campus), Zhuhai, China. [5]State Key Laboratory of Optoelectronic Materials and Technologies, Sun Yat-Sen University (Guangzhou Campus), Guangzhou, China. ✉e-mail: B.D.Gerardot@hw.ac.uk

**Fig. 1 | Device and electronic properties of R-stacked bilayer WSe₂. a** Side and top view of AB and BA stacking configurations. **b** Schematic representation of the dual-gated R-WSe₂ device with ferroelectric domains. **c** Reflectance contrast spectra of monolayer and bilayer R-WSe₂, highlighting the low-energy (LE) and high-energy (HE) exciton peaks in the bilayer. **d** Schematic illustration of how the atomic registry and intrinsic polarization in R-stacking TMDs lifts layer degeneracy, creating type-II band alignment. **e** 2D map of the LE exciton reflectance contrast (indicated as the dashed brown line in (**c**)) as a function of top ($V_{tg}$) and bottom ($V_{bg}$) gate voltages. The dashed black lines indicate boundaries between different electronic regimes (I–VI) explored in this work.

demonstrated robust polarization, domain formation, and excitonic signatures of ferroelectric order in R-stacked MoS₂ and WSe₂ homobilayers[7–11]. Moreover, the combination of ferroelectric, piezo-electric, and strain fields at AB/BA domain boundaries and intersections can create confinement potentials for excitons, effectively forming arrays of quantum-dot-like states under appropriate conditions[12], opening opportunities to harness switchable ferroelectric materials for quantum optoelectronics.

While the intrinsic ferroelectricity of R-stacked homobilayers represents a striking phenomenon in its own right, it also plays a defining role in the emergent physics of twisted bilayers. Among homobilayer platforms, small-angle twisted bilayer WSe₂ (tWSe₂) has recently emerged as a system of profound interest[13–17]. The discovery of superconductivity in tWSe₂ has placed it as a key material to explore correlated physics beyond the graphene family[16,17]. A crucial feature of tWSe₂ at near-zero twist angles is that the lattice reconstructs into a periodic superlattice of nanometer-scale ferroelectric domains, providing the building blocks for moiré superlattices with flat bands and correlated states[18,19]. In the 0-degree (untwisted) limit, which is the focus of our study, these domains further relax into large, micrometer-scale regions (see schematic in Fig. 1b). The spontaneous polarization of these nanometer-scale domains and the electrostatic fields at their boundaries imprint a highly nontrivial potential landscape that shapes the moiré flat bands. Recent theoretical work suggests that the interplay of intrinsic ferroelectricity with strain-induced piezoelectricity governs both the topology and the correlated electronic phases in tWSe₂ and related materials[20]. This highlights a critical point: a quantitative and comprehensive understanding of the untwisted R-stacked bilayer WSe₂—including the precise band alignment, the magnitude of the built-in polarization field, and its tunability—is an indispensable prerequisite for understanding the complex interplay of correlation, topology, and superconductivity observed in its twisted counterpart. However, these fundamental properties of R-WSe₂ remain largely unexplored.

In this work, we address this critical knowledge gap by performing a systematic low-temperature optical spectroscopy study on a dual-gated, untwisted R-stacked WSe₂ bilayer. We leverage the sensitivity of excitons and exciton-polarons to the local electronic environment to map the material's fundamental properties with high precision. We first uncover a distinct asymmetry in the system's response to electron

and hole doping, which unambiguously establishes a type-II band alignment at the K-valleys and identifies the conduction and valence band edges to be at the Λ and K points, respectively. By applying a small external electric field and tracking the doping-dependent evolution of excitonic features, combined with the observation of inter-layer exciton hybridization, we provide direct optical evidence for the coexistence of AB and BA ferroelectric domains.

Furthermore, we employ exciton-polarons as a sensitive probe to quantitatively determine the magnitude of the intrinsic built-in electric field arising from spontaneous polarization. Strikingly, we demonstrate precise control over the band structure, observing a layer symmetric valence band maximum (VBM) switch under a strong electric field, which we attribute to field-induced ferroelectric domain switching. Our work provides a complete electronic band picture and a set of crucial experimental parameters—including the band-gap difference ($\delta$), interlayer potential ($\phi_0$), and valence band offset ($\Delta_v$)—that lay the essential foundation for understanding the emergent correlated and superconducting phases in twisted WSe₂ systems.

## RESULTS

### Charge- and field-tunable electronic structure of R-stacked WSe₂

Our experimental platform is an electrically contacted 0° R-stacked bilayer WSe₂ fully encapsulated in ~40 nm thick hexagonal boron nitride (hBN) with graphite top and bottom gates (Fig. 1b, Fig. S1), enabling a dual-gate geometry that allows independent tuning of carrier density ($n$) and perpendicular displacement field ($D$). The bilayer was fabricated using the tear-and-stack technique[21], which reliably produces micrometer-scale ferroelectric domains (Fig. S2). Additional control experiments confirm that these ferroelectric domains form robustly at the interface, even in the presence of encapsulation bubbles (see Fig. S3). To probe the electronic structure of this device, we employ low-temperature optical reflectance contrast spectroscopy at 4 K. This technique sensitively probes interband transitions, providing direct access to excitonic resonances that reflect the underlying band structure. As shown in Fig. 1c, while monolayer WSe₂ exhibits a single neutral exciton resonance, the R-stacked bilayer displays two distinct neutral exciton peaks. We label these the low-energy (LE) exciton at 1.6757 eV and the high-energy (HE) exciton at 1.6899 eV (see Fig. S4 for more data from other positions), which correspond to intralayer transitions in the two non-equivalent WSe₂

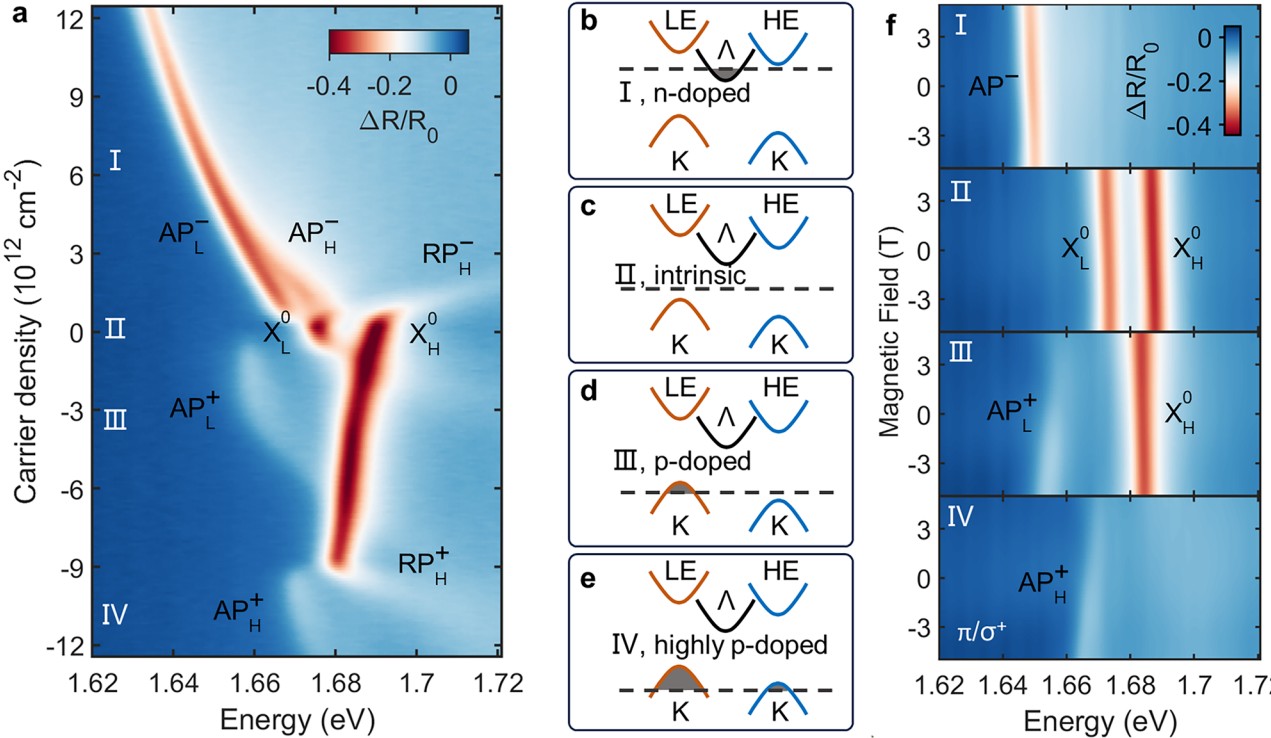

**Fig. 2 | Asymmetric carrier doping and band alignment. a** Doping-dependent reflectance contrast at $D = 0$. Attractive (AP) and repulsive (RP) polarons are formed with doping ($^-$ for electrons, $^+$ for holes, $_L$ for low energy and $_H$ for high energy exciton). **b–e** Schematics of the band filling sequence for electron doping (**b**),

intrinsic (**c**), light hole doping (**d**), and heavy hole doping (**e**), illustrating the type-II band alignment at the K-point and the different valley locations for electrons ($\Lambda$) and holes (K). **f** Magnetic field dependent reflectance contrast under different doping conditions with linear polarization excitation and $\sigma^+$ detection.

layers[11]. The measured energy difference between these excitons, which reflects the intrinsic band-gap difference $\delta$ between the two layers, is 14.2 meV. Full-width at half-maximum (FWHM) values of 7.2 meV and 9.8 meV are observed for the LE and HE peaks, respectively.

The electronic asymmetry between the AB and BA stacking configurations in R-stacked WSe$_2$ (Fig. 1a) lifts the layer degeneracy and produces a type-II band alignment (Fig. 1d). This structure can be quantitatively described by two key contributions[10,22,23]:

1. **Direct band-gap difference ($\delta$):** The in-plane lateral displacement of one layer relative to the other places tungsten and selenium atoms in non-equivalent local registries (e.g., metal over chalcogen vs. hollow site), resulting in slightly different intrinsic band gaps for the two layers.

2. **Interlayer potential ($\phi_0$):** The interlayer coupling in the AB/BA configuration generates a spontaneous ferroelectric polarization, which creates a built-in electrostatic potential between the layers.

Layer-dependent hopping processes between conduction and valence bands (including higher-energy states) further renormalize the effective band edges. We account for this complexity with an asymmetric interlayer coupling coefficient $\alpha$ that modifies the direct band-gap difference $\delta$. The combination of these effects determines the conduction- and valence-band offsets between the two layers, which can be expressed as[23]:

$$\Delta_c = \alpha\delta + e\phi_0, \tag{1}$$

$$\Delta_v = (\alpha + 1)\delta + e\phi_0. \tag{2}$$

These band offsets provide a useful framework, but their precise values and tunability must be determined experimentally. To this end,

we measure the differential reflectivity while sweeping both top ($V_{tg}$) and bottom ($V_{bg}$) gate voltages from $-12$ V to 12 V. As a representative probe, we track the reflectance contrast at the LE exciton energy (1.6757 eV), which captures the interplay between carrier doping, band filling, and the underlying ferroelectric order of the bilayer. The resulting 2D map (Fig. 1e) reveals a rich electronic landscape with multiple sharp boundaries that demarcate regions of distinct electronic character (labeled I–VI). In the following sections, we systematically deconstruct this landscape to unveil the underlying physics.

**Asymmetric carrier doping and band alignment at zero field**

To understand the origin of the complex excitonic landscape in Fig. 1e, we first investigate the intrinsic band structure by studying the system's response to carrier doping at zero displacement field ($D = 0$). The voltage scanning path is shown in Fig. S5. Figure 2a shows the doping-dependent reflectance contrast map. In the intrinsic region ($n \approx 0$), two prominent neutral exciton peaks, corresponding to the LE and HE transitions, are clearly visible. We label these as $X_L^0$ and $X_H^0$, respectively.

Upon electron doping ($n > 0$), the spectra evolve significantly. The neutral exciton peaks $X_L^0$ and $X_H^0$ blueshift and form repulsive polarons, while two new attractive polaron branches ($AP_L^-$ and $AP_H^-$) emerge at lower energies[24]. As the electron concentration increases, both attractive polarons exhibit a pronounced redshift. Notably, $AP_H^-$ redshifts more rapidly than $AP_L^-$, eventually merging with it at a density of $n > 3.0 \times 10^{12}$ cm$^{-2}$. A redshift of attractive polarons is a key signature that the doped carriers do not occupy the K-valley states associated with the excitons[25,26]. This observation is consistent with theoretical predictions that the conduction band minimum in this system is located at the $\Lambda$ valley[27] (also known as Q valley in literature[28,29]). This conclusion is further corroborated by g-factor measurements (Fig. 2f, Figs. S6, S7), where the g-factors of $AP^-$ are found to be identical to those of the neutral excitons, confirming that electrons are not doped

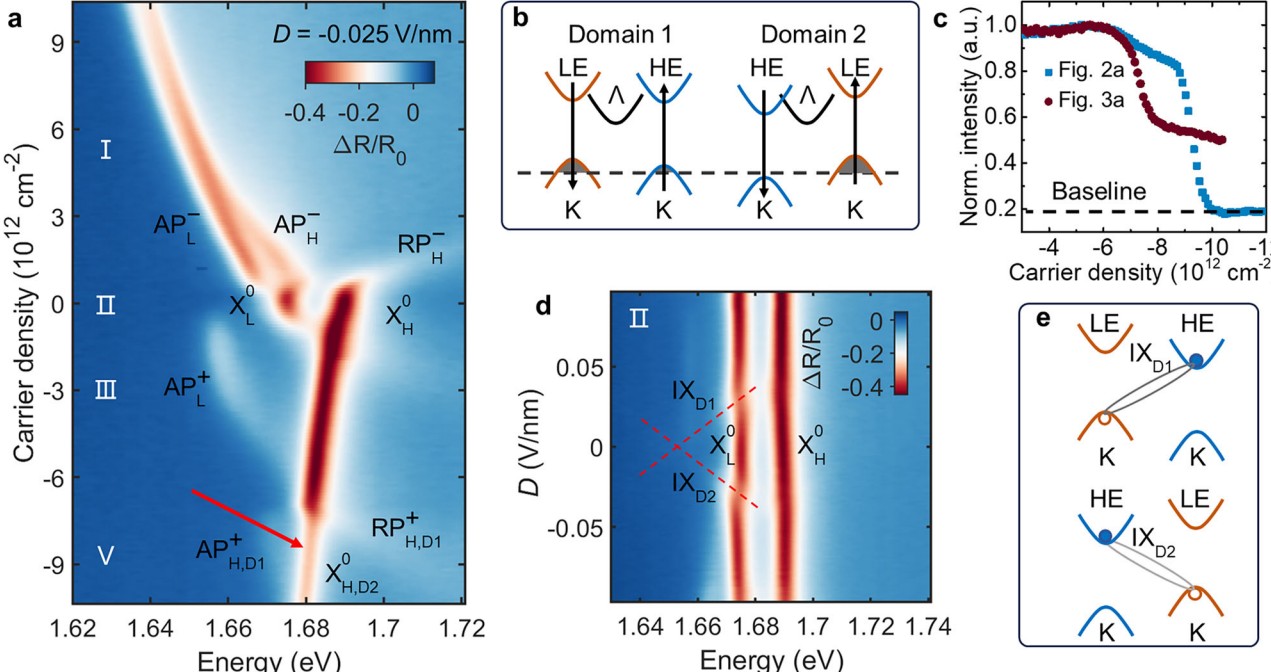

**Fig. 3 | Optical identification of AB/BA mixed ferroelectric domains.**
**a** Reflectance map under a small field of $D = -0.025$ V nm$^{-1}$. **b** Schematic showing the opposite response of AB and BA domains to an external electric field. **c** Hole-doping dependence of the HE exciton oscillator strength, which plateaus at -50%

for (**a**), evidencing the presence of two domains. The oscillation strength is normalized to its maximum intensity in the plot range. **d** Hybridization of interlayer exciton (IX) and intralayer exciton, the different slopes of IX reveal the coexistence of AB/BA domains. **e** Energy level diagram explaining the origin of the two IX peaks.

into the K-valley. The g-factor for $AP^-$, $X_L^0$ and $X_H^0$ are $-5.04 \pm 0.03$, $-4.89 \pm 0.04$ and $-5.26 \pm 0.08$, respectively. The different redshift rates of the two polarons suggest that the electrons in the $\Lambda$ valley are not symmetrically distributed between the layers, but are instead polarized towards the HE layer, likely due to the built-in electric field.

The more complex response under hole doping ($n < 0$) reveals the staggered nature of the valence bands at the K-points. Initially, as holes are introduced, the $X_L^0$ peak loses oscillator strength and forms a repulsive polaron ($RP_L^+$) and a corresponding attractive polaron ($AP_L^+$). The $AP_L^+$ branch blueshifts with increasing hole density - a hallmark of phase-space filling as carriers are doped directly into the associated K-valley band. This is confirmed by g-factor measurements (Fig. 2f), which show a much larger g-factor for $AP_L^+$ ($24.39 \pm 1.70$) than for the neutral exciton[25]. During this initial doping stage, the $X_H^0$ peak maintains its oscillator strength while exhibiting a slight redshift, likely due to changes in the effective dielectric environment. As hole doping is further increased, a second threshold is crossed at $n \approx 9.1 \times 10^{12}$ cm$^{-2}$. Beyond this point, the $X_H^0$ peak also begins to quench, forming $RP_H^+$ and $AP_H^+$. This sequential filling process provides direct evidence that the K-valley valence bands of the two layers are staggered in a type-II alignment.

From Fig. 2a, we can extract the binding energies of the various polaron states. For the electron-doped side, the binding energies of $AP_L^-$ and $AP_H^-$ are 8.7 meV and 16.4 meV, respectively. We attribute this significant difference primarily to the built-in electric field, which localizes the doped electrons closer to the HE layer, thereby enhancing the exciton-carrier interaction. For the hole-doped side, the binding energies for $AP_L^+$ and $AP_H^+$ are 17.6 meV and 10.5 meV. This disparity stems from varied dielectric screening, as the total carrier concentration is substantially higher when $AP_H^+$ forms compared to when $AP_L^+$ forms.

The schematics in Fig. 2b–e summarize the band alignment and carrier filling sequence derived from our doping-dependent spectroscopy. This comprehensive picture allows us to directly map the electronic configurations to the distinct regions observed in the gate

map of Fig. 1e, with regions I-IV corresponding to the sequential filling of the hole and electron bands as described above.

## Optical signatures of mixed AB/BA ferroelectric domains

To further validate the influence of ferroelectric domains on the band structure, we apply a small out-of-plane electric field of $D = -0.025$ V nm$^{-1}$. Since the AB and BA domains possess opposite stacking structures (see Fig. 1a), the response of their respective LE and HE layers to the external field will also be inverted. Figure 3a displays the reflectance spectrum as a function of doping concentration under this field. On the electron-doped side, the spectrum remains largely unchanged compared to the zero-field case. The low-concentration hole-doped regime also shows similar behavior to that shown in Fig. 2a, where only the LE layer is populated.

However, significant differences emerge once the hole concentration is high enough to begin populating the HE layer. Specifically, two key changes are observed. First, the doping threshold to populate the HE layer is reduced from $n \approx 9.1 \times 10^{12}$ cm$^{-2}$ (at $D = 0$) to $n \approx 7.3 \times 10^{12}$ cm$^{-2}$. Second, after the formation of the attractive polaron $AP_{H,D1}^+$, and repulsive polaron $RP_{H,D1}^+$, where the D1 subscript is used to label Domain 1, the oscillator strength of the neutral exciton $X_H^0$ does not quench to zero as it did in Fig. 2a. Instead, it maintains a considerable intensity.

These phenomena can be understood by considering the effect of the applied field on the two different domain types within the optical focus, as depicted in the schematic in Fig. 3b. In Domain 1, the external field raises the valence band of the HE layer, making it easier to populate and thus lowering the doping threshold. Conversely, in Domain 2, the field lowers the HE layer's valence band, requiring a larger change in the Fermi level to achieve doping into the layer. This explains why the $X_H^0$ oscillator strength persists even after doping commences in the HE layer of Domain 1. We plot the intensity of $X_H^0$ as a function of doping in Fig. 3c. At zero field, the two domains are degenerate, and the $X_H^0$ intensity drops sharply to a baseline signal as the HE layer is doped. Under a field of $D = -0.025$ V nm$^{-1}$, the intensity

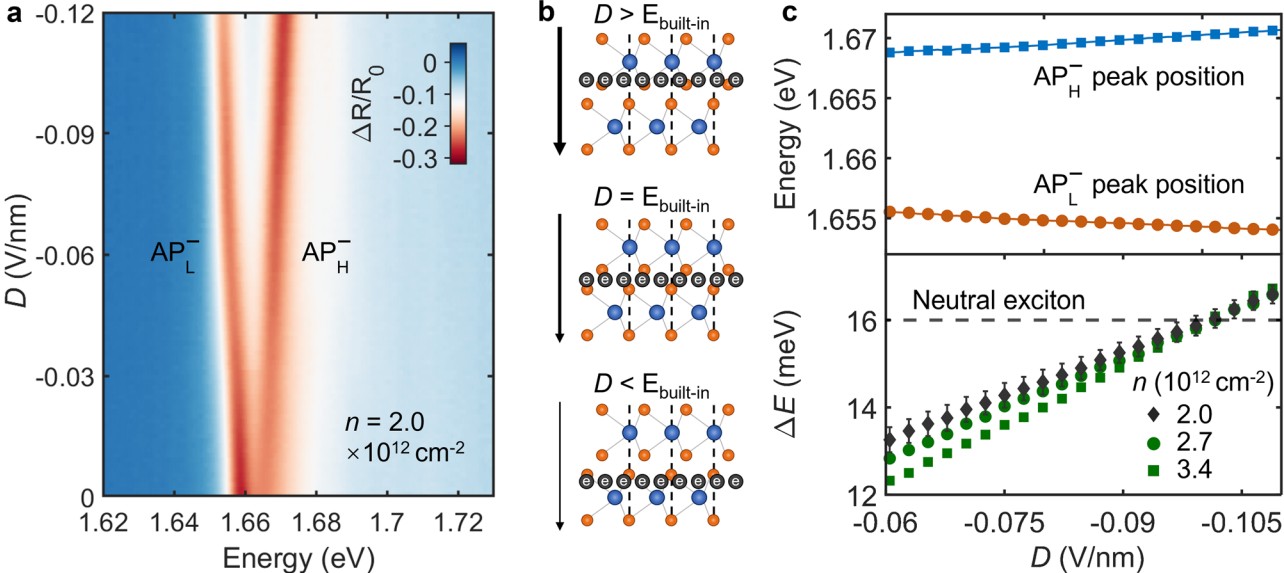

**Fig. 4 | Quantitative measurement of the built-in ferroelectric field.**
**a** Reflectance map at a fixed electron density of $n = 2.0 \times 10^{12}$ cm$^{-2}$ as a function of displacement field, showing the evolution of attractive polaron energies.
**b** Schematic illustrating how the external field tunes the electron distribution between layers. **c** Peak energies of the attractive polarons (AP$_L^-$, AP$_H^-$) and their

energy difference ($\Delta E$) versus displacement field with different doping levels. The built-in field is identified at $D \approx 0.102$ V/nm, where the polaron energy splitting matches that of the neutral excitons (indicated with a dashed black line). The error bars represent the standard errors of the fitted peak center positions.

drops as Domain 1 is doped but then stabilizes at a plateau. Based on the relative intensity change, we can estimate that the laser spot covers an area with an AB to BA domain ratio of approximately 1:1 for this particular sample position. See Figs. S8, S9 for data at another spatial position, which has only a single ferroelectric domain. We note that the data from the mixed-domain region not only provides a powerful, simultaneous demonstration of the opposite AB/BA response but is also more representative of typical R-stacked samples where domain sizes are often smaller than the optical spot.

The presence of mixed domains is further confirmed by the hybridization of interlayer excitons (IX) with intralayer excitons under an applied field, shown in Fig. 3d. An interlayer exciton species is observed to hybridize with an intralayer exciton at an electric field of $D \approx 0.02$ V nm$^{-1}$. Crucially, this anti-crossing behavior exists for both positive and negative polarities of the applied field, which is a definitive signature of the coexistence of both AB and BA domains, as they exhibit opposite Stark shifts. The corresponding energy level schematic is shown in Fig. 3e. At the same time, we note that due to the coupling between interlayer and intralayer excitons, the actual energy offset between the two layers is slightly larger than that at zero displacement field. By extracting the intralayer exciton energy splitting away from the hybridization region and averaging around $D \approx -0.07$ V nm$^{-1}$, we obtain a more accurate estimate of $\delta \approx 16.0$ meV.

## Quantifying the ferroelectric built-in field

Having confirmed the existence of mixed domains, we next seek to determine the precise magnitude of the built-in electric field. As established earlier, the intrinsic field polarizes the doped Λ-valley electrons, pulling them closer to the HE layer. By applying an external out-of-plane electric field, we can counteract this effect and, by finding the point of cancellation, directly measure the internal field's strength.

To avoid averaging of this effect over both domains, we perform measurements on a sample area identified as having a single ferroelectric domain within our optical focus spot (see Figs. S8–S10). We apply a constant electron doping concentration of $n = 2.0 \times 10^{12}$ cm$^{-2}$ and sweep the external displacement field $D$. The key insight is that

when the external field exactly cancels the internal field, the Λ-valley electrons are symmetrically distributed between the two layers. At this symmetric point, the Λ electron gas interacts identically with both the HE and LE excitons, leading to an equal energy redshift for both polaron states. Consequently, the energy difference between the attractive polarons, AP$_L^-$ and AP$_H^-$, becomes equal to the energy difference of the neutral excitons, X$_L^0$ and X$_H^0$.

Figure 4a shows the evolution of the AP$_L^-$ and AP$_H^-$ peaks as a function of the applied field $D$ (see Fig. S11 for the whole scan range). As the field strength increases, the AP$_L^-$ peak redshifts while the AP$_H^-$ peak blueshifts. This behavior is a clear signature of the electron wavefunction being transferred from the vicinity of the HE layer toward the LE layer, as illustrated schematically in Fig. 4b.

We extract the peak positions and plot the energy difference between them, $\Delta E$, as a function of the applied field in Fig. 4c. The energy difference $\Delta E$ exhibits a nearly linear dependence on the field. The critical point is where this energy difference matches that of the neutral excitons (the dashed line in the plot). This condition is met at an applied field of $D \approx 0.102$ V nm$^{-1}$. This value directly corresponds to the magnitude of the intrinsic built-in electric field, $E_{\text{built}-\text{in}}$. We find that this field strength is nearly independent of the doping concentration. Field-dependent $\Delta E$ for other doping levels is also summarized in Fig. 4c, which yields the same built-in electric field value.

Based on this measured field, we can calculate the corresponding interlayer potential, $\phi_0$, using the relation $\phi_0 = E_{\text{built}-\text{in}} \times d_0$, where $d_0$ is the interlayer distance. Using a typical value of $d_0 \approx 0.65$ nm for bilayer TMDs, we obtain:

$$\phi_0 = 0.102 \, V \, nm^{-1} \times 0.65 \, nm = 66.3 \, mV \qquad (3)$$

This value for the interlayer potential is comparable in magnitude to earlier measurements on R-stacked WSe$_2$ using alternative techniques[30], as well as to values reported in other R-stacked TMDs such as MoS$_2$[10], establishing a quantitative measure of the spontaneous polarization in bilayer WSe$_2$.

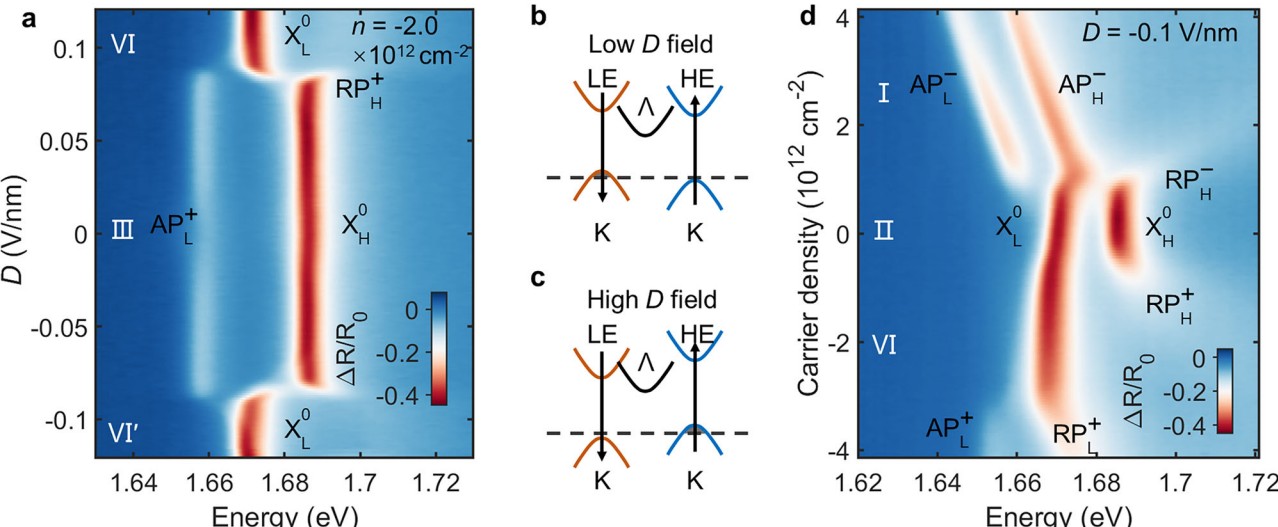

**Fig. 5 | Highly tunable valence band and domain switching. a** Reflectance map in the hole-doped regime versus displacement field, showing an abrupt and symmetric spectral shift indicative of valence band maximum switching and domain switching. **b, c** Schematics illustrating the valence band maximum switching under a strong electric field. **d** Doping dependence measured after the switching event, confirming the inverted band alignment.

## Highly tunable valence band and domain switching

Next, we manipulate the relative positions of the LE and HE layer valence band maxima by applying an electric field to determine the valence band offset, $\Delta_v$. To begin, we fix the hole doping concentration at $n = 2.0 \times 10^{12}$ cm$^{-2}$, a density at which only the LE layer's valence band is populated at zero field. As we gradually increase the external field, the spectrum undergoes an abrupt transformation at a critical field of $D \approx 0.086$ V nm$^{-1}$, as shown in Fig. 5a. At this point, the $X_H^0$ oscillator strength suddenly decreases as it forms a repulsive polaron ($RP_H^+$), while the $AP_L^+$ peak vanishes, accompanied by the re-emergence of the neutral $X_L^0$ exciton.

This sudden spectral change indicates that a crossover has occurred, and the HE layer's valence band has become the new valence band maximum (VBM). The switching behavior is further supported by doping dependence measurement near the switching field (Fig. S12). The underlying band structure evolution is depicted schematically in Fig. 5b, c. Due to the spectral overlap between the newly formed $AP_H^+$ and the reappeared $X_L^0$, we cannot resolve them as distinct peaks. From this VBM switching experiment, we determine the valence band offset to be:

$$\begin{aligned}
\Delta_v &= e \times D_{switch} \times d_0 \\
&= e \times 0.086\ V/nm \times 0.65\ nm \\
&= 55.9\ meV
\end{aligned} \tag{4}$$

Surprisingly, we observe a similar VBM switching event at the symmetrically opposite field strength of $D \approx -0.086$ V nm$^{-1}$. This is unexpected, as a negative field should theoretically drive the LE and HE valence bands further apart. This observation strongly implies that the domain itself has undergone a ferroelectric switch at a field strength below $-0.086$ V nm$^{-1}$, converting its stacking configuration to the opposite type (e.g., AB → BA). The system then behaves as a new domain with a reversed built-in field. This dynamic domain switching is independently verified through IX photoluminescence experiments (Fig. S13).

To confirm this band structure reconfiguration, we fix the field in the VBM switched state at $D = -0.100$ V nm$^{-1}$ and study the doping dependence (Fig. 5d). On the electron-doped side, because the external field is now close in magnitude to the built-in field, electrons are more evenly distributed than in the scenario of Fig. 3a, causing the $AP_L^-$ and $AP_H^-$ peaks to redshift nearly in parallel. On the hole-doped side,

the filling sequence is now inverted: the $X_H^0$ peak quenches first, forming $RP_H^+$. Although the corresponding $AP_H^+$ is obscured by the $X_L^0$ peak, its presence can be inferred from the change in the $X_L^0$ lineshape. At higher hole concentrations, the LE layer also begins to fill, forming $AP_L^+$ and $RP_L^+$.

## Discussion

In this work, we have performed a comprehensive optical spectroscopic study of R-stacked bilayer WSe$_2$, systematically unraveling its fundamental electronic and ferroelectric properties. Our key findings are threefold. First, we have established its asymmetric band structure, determining that the lowest-energy electron and hole states reside in the Λ and K valleys, respectively. Second, we have provided direct optical identification of coexisting AB and BA ferroelectric domains and presented an all-optical method to quantify the intrinsic built-in field. Third, we have demonstrated a highly tunable valence band maximum in ferroelectric bilayer WSe$_2$, which may enable new opportunities with ferroelectric tunable moiré quantum systems, such as WS$_2$/bilayer WSe$_2$ or MoSe$_2$/bilayer WSe$_2$ platforms.

Our experiments yield quantitative values for the key parameters governing the band alignment: the direct band-gap difference $\delta = 16.0$ meV, the interlayer potential energy $e\phi_0 = 66.3$ meV, and the valence band offset $\Delta_v = 55.9$ meV. These values allow us to estimate the coefficient $\alpha$, which incorporates the effects of non-equivalent atomic registries and asymmetric interlayer coupling, using the relation derived from the theoretical model: $\alpha = (\Delta_v - e\phi_0)/\delta - 1$. Substituting our experimental values gives $\alpha = (55.9\ \text{meV} - 66.3\ \text{meV})/16.0\ \text{meV} - 1 = -1.65$. With $\alpha$, we can also calculate the conduction band offset $\Delta_c = \alpha\delta + \phi_0 = -1.65 \times 16.0\ \text{meV} + 66.3\ \text{meV} = 39.9$ meV.

These results provide the essential physical picture and experimental parameters that are indispensable for building accurate theoretical models of twisted bilayer WSe$_2$. A thorough understanding of the parent compound is a critical prerequisite for decoding the complex interplay of correlations, topology, and superconductivity in the moiré derivatives. Our work thus lays a vital foundation for future exploration and manipulation of quantum phases in this fascinating family of 2D materials and paves the way for new ferroelectric and optoelectronic devices. Specifically, the electric-field control over coexisting ferroelectric domains, which we demonstrate, is a key mechanism for developing ultra-low power sliding ferroelectric memories. As recent work has highlighted, these systems show great

promise for applications such as neuromorphic computing (e.g., as artificial synapses)[31] and sequential logic-in-memory circuits[20]. Furthermore, coupling these controllable ferroelectric domains with other intrinsic 2D properties, such as the spin-valley locking in $WSe_2$, could enable spintronic devices like spin field-effect transistors or spin logic gates. This platform also offers a new route to electrically switch or tune quantum phases, such as superconductivity, in related moiré twisted bilayers.

## Methods

### Device fabrication

High-quality bulk crystals of $WSe_2$ and hBN were used. Monolayer $WSe_2$ was mechanically exfoliated onto a $Si/SiO_2$ substrate. An R-stacked bilayer was created using the "tear-and-stack" method under an optical microscope[21]. The heterostructure was assembled using a standard dry-transfer technique, resulting in the bilayer $WSe_2$ being encapsulated between two hBN flakes, with thin graphite flakes serving as top and bottom gates. A topmost thick hBN ($\approx$50 nm) flake was used to cover and protect the whole sample. Pre-patterned electrical contacts were defined by laser lithography followed by evaporation of Cr/Au (5 nm/50 nm).

### Optical spectroscopy

All measurements were performed in a closed-cycle optical cryostat (Attodry 1000) at a base temperature of 4 K. Reflectance contrast spectra were acquired using a home-built microscope setup. White light from a tungsten-halogen lamp was focused onto the sample using an apochromatic objective (NA = 0.82), and the reflected light was collected and analyzed by a spectrometer equipped with a liquid-nitrogen-cooled CCD camera. Photoluminescence measurements were performed using a 532 nm continuous wave laser with a power of 30 nW.

## Data availability

The data that support the findings of the study are included in the main text and supplementary information files. Raw data can be obtained from the corresponding author upon request. Source data are provided with this paper.

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

## Acknowledgments

This work was supported by the EPSRC (Grants No. EP/P029892/1 and No. EP/Y026284/1 to B. D. G.). Z. L. is supported by a Marie Skłodowska-Curie Individual Fellowship (No. 101208583). M. B.-G. is supported by a Royal Society University Research Fellowship. B. D. G. is supported by a Chair in Emerging Technology from the Royal Academy of Engineering. H.Y. acknowledges support by NSFC under grant No. 12274477. K. W. and T. T. acknowledge the support from the JSPS KAKENHI (Grants No. 21H05233 and No. 23H02052), the CREST (JPMJCR24A5), JST, and World Premier International Research Center Initiative (WPI), MEXT, Japan.

## Author contributions

Z.L. performed the optical measurements with the assistance of P.T. and T.I. Z.L. fabricated the sample. Z.L. and G.K. carried out the Kelvin probe

force microscopy measurements. Z.L., P.T., H.Y., M.B.-G., and B.D.G. analyzed the data. T.T. and K.W. grew the hBN crystals. B.D.G. supervised the project. Z.L. and B.D.G. wrote the manuscript with input from all authors.

## Competing interests

The authors declare no competing interests.
