## [Transparent Peer Review file · Nature Communications]

Highly tunable band structure in ferroelectric R-stacked bilayer WSe₂

Corresponding Author: Professor Brian Gerardot

Version 0:

Reviewer comments:

Reviewer #1

(Remarks to the Author)

The authors present a detailed investigation of untwisted R-stacked WSe₂ bilayers and successfully identify the coexistence of AB and BA ferroelectric domains. The results are scientifically intriguing and contribute meaningfully to the field of twistrionics. However, I believe the current version of the manuscript requires further clarification and supporting data before it can be considered for publication. My concerns are outlined below:

1. The reliability of the band alignment analysis critically depends on the quality of the R-stacked bilayer. While the authors mention the use of the “tear-and-stack” method, the manuscript lacks sufficient characterization data to confirm sample uniformity. Given the nature of the stacking process, the presence of bubbles or other inhomogeneities is likely. Therefore, comprehensive data such as AFM topography across the entire sample, PL/Raman mapping, and representative spectra should be provided to validate the sample quality.
2. As illustrated in Fig. 1b, domains induced by stacking are expected to exhibit symmetry or regular patterns. However, the KPFM data shown in Fig. S1 does not clearly demonstrate such regularity. The authors are encouraged to explain this discrepancy and discuss possible reasons for the observed domain distribution.
3. It is unclear which specific region of the sample was selected for the measurements presented in Fig. S1. Can identical optical properties be consistently observed across all domains that exhibit similar contrast in Fig. S1? If so, please provide supporting evidence. Furthermore, is the data reproducible even within the same domain? Clarification and corresponding data are requested. Since multiple domains are visible, a more detailed justification for the choice of measurement location would significantly enhance the credibility of the experimental results.
4. The abstract mentions “establishing key parameters to understand twisted bilayers and enabling new ferroelectric and excitonic device opportunities,” and a similar statement appears in the conclusion. To strengthen the impact of the manuscript, the authors should elaborate on how their findings could be applied in practical device contexts, with specific examples or mechanisms.

In summary, while the manuscript focuses on deep scientific insights, additional explanation and supporting data are necessary to improve its accessibility and impact. In particular, given the experimental complexity, the quality of the sample plays a crucial role. I strongly recommend the inclusion of more thorough sample characterization to ensure the reliability of the reported results.

Reviewer #2

(Remarks to the Author)

The manuscript entitled “Highly tunable band structure in ferroelectric R-stacked bilayer WSe₂” by Zhe Li et al. presents interesting results on the analysis of key parameters affecting the electronic band structure of untwisted R-stacked bilayer WSe₂ using optical measurements. The device scheme involving a dielectric-sandwiched R-stacked bilayer WSe₂ to understand doping via the same polarity gating and ferroelectric built-in potential via opposite polarity gating is a novel approach to understanding the complex type-II band alignment in R-stacked bilayer WSe₂. Their experimental results and the proposed model for understanding the band-gap difference, interlayer potential, and band offset in R-stacked bilayer WSe₂ are informative and convincing. Thus, I suggest minor revisions before this manuscript can be considered suitable for publication in Nature Communications.

- i) The authors present KPFM data in Fig. S1 to confirm that the device stacking is R with either BA or AB stacking order in bilayer WSe₂. However, the surface potential values for both stackings are positive. Could the authors explain why both stackings show positive values, given that the built-in potential for BA and AB R-stacked WSe₂ should be opposite? Also, is KPFM data alone sufficient to prove that the WSe₂ bilayer is untwisted R-stacked?
- ii) Given the broad readership of Nature Communications, the authors should provide more detailed information on key experimental parameters and the calculation process. Specifically, the manuscript should present detailed information on the reflectance contrast ($\Delta R/R_0$ and normalized $\Delta R/R_0$). Please also include the calculation process for evaluating the doping concentration in R-stacked bilayer WSe₂ using dual gating, as well as the calculation for the g-factor. Additionally, the g-factor for AP- is slightly different from that of the neutral excitons. Can these values be treated as identical, as mentioned on page 3?
- iii) On page 6, the authors suggest a method for measuring the internal potential in R-stacked WSe₂ by sweeping the external electric field D. How can the internal potential be determined by finding the external electric field D value that results in an equal energy difference between the attractive polarons and the neutral excitons of both layers?
- iv) Regarding ferroelectric switching in R-stacked bilayer WSe₂, why does only the p-doped R-stacked bilayer WSe₂ show switching behavior in contrast to the n-doped R-stacked bilayer WSe₂?
- v) In Fig. 1(b), the device schematic for untwisted R-stacked WSe₂ with two domains could mislead readers since it looks like a twisted R-stacked WSe₂ with a moiré structure. Please clarify that the orange or white domains represent opposite stacking orders, not a moiré structure.
- vi) Physical quantities should be italicized. For example, in Fig. 1(e), V should be italicized. Please correct any typos throughout the manuscript.
- vii) While I agree with the authors' claim that untwisted R-stacked bilayer WSe₂ exhibits LE and HE type-II band structures regardless of stacking order, it seems that optical measurements cannot determine which layer is LE or HE. Is there an efficient method to distinguish the LE layer from the HE layer in untwisted R-stacked bilayer WSe₂?

Reviewer #3

(Remarks to the Author)

The manuscript entitled "Highly tunable band structure in ferroelectric R-stacked bilayer WSe₂" by Li et al. reveals the exciton complex in the R-stacked bilayer WSe₂. The data quality is high standard. The switching between the AB and BA stacks was detected in the photoluminescence spectra. Nevertheless, there are several concerns that the authors should address before it can be recommended for publication in Nature Communications.

1. The size of the AB and BA domains is several micrometers. It indicates that the authors can investigate in AB and BA domains. The data shown in Figs. S4 and S5 were claimed in a single domain. Why did the author choose the data from mixed domains to analyze in the main manuscript? This makes the data analysis more complicated, and it is difficult for the reader to follow.
2. Compare the data of Fig. 3a (multidomain), Fig. S4b (single domain), and Fig. S5 (single domain); it is difficult to see the difference. Could the author provide more analysis to make the claim of multidomain more convincing? Could the authors provide the spectra at that specific multidomain and compare them with the corresponding spectra of the single domain?
3. Could the authors provide a similar analysis of Fig. 3c for Fig. S4 and Fig. S5?
4. In Figure 3b, the appearance of the Q valley is the lowest energy point. Is the indirect exciton from the K-K valley in Fig. 3e correct? Is it possible from the Q-K valley?
5. Could the author provide the data of AB and BA PL, as shown in Fig. 1c? It is interesting to see how it is different.
6. Could the author provide the data for the positive D field, as in Fig. 4a for the negative D field?

In addition, I have other comments:

- The introduction part is not convincing to me. The relaxation at the micrometer AB and BA deteriorates the flat band. The citation in this part is for STM, with AB and BA at the nanometer scale. That part of the introduction should be revised.
- Fig. 1a: Adding a top view of the structure will be useful.
- Figure 1d: Exaggerate bandgap differences, make the figure more informative.
- Single spectrum (e.g., Fig. 1c) with different regions?

Version 1:

Reviewer comments:

Reviewer #1

(Remarks to the Author)

Most of my questions have been addressed with fairly thorough answers. However, I have found a serious problem in the interpretation of the experimental results. Specifically, the two different domains, which are the core of this paper, must be clearly distinguished. The domains were clearly identified in Figure S2 using KPFM, and that part is relatively clear. However, the optical analysis was conducted on an entirely different sample, not the one confirmed by KPFM. Figure R8

presents such a sample image.

The problem lies in the method used to distinguish the two different domains. The authors attempted to differentiate the domains by using reflectance data, marking the positions of the two domains with white and red stars. Yet, based on the presented results, it is difficult to objectively distinguish the domains. In other words, there is no direct evidence that the positions marked by the red and white stars correspond to different domains.

Figure R8(b) shows changes in the reflectance contrast spectra, but these signals can also vary depending on the sample quality (e.g., interface contaminations, bubbles, wrinkles). In fact, Figure S1 BF/DF images reveal that many bubbles are present in the sample. Figure S1 (d) marks the positions of bubbles, and the authors argue in Figure S3 that measurements were taken at clean locations. However, microscopic bubbles or contamination cannot be identified solely with an optical microscope. This means that there could be subtle impurities at the positions marked by the red and white stars that are not distinguishable optically.

Therefore, more definitive experimental evidence is required. Unless direct proof is provided that the key results discussed in the main text were indeed measured from different domains, the experimental results cannot be considered reliable at all.

Reviewer #2

(Remarks to the Author)

The revised manuscript entitled "Highly tunable band structure in ferroelectric R-stacked bilayer WSe₂" by Zhe Li et al. have satisfactorily addressed all my previous comments and made corresponding changes to the manuscript and supplementary information. Now I recommend the study "Highly tunable band structure in ferroelectric R-stacked bilayer WSe₂" for publication in Nature communications.

Reviewer #3

(Remarks to the Author)

My concerns in the original manuscript were nicely resolved by the authors in the revised version. Nevertheless, before I can accept the manuscript for publication in Nature Communications, the authors should incorporate the additional data from the reviewers' response into the revised version, at least in the supplementary information. For example, the data in Figures R5 and R6. These will help readers better understand the quality of the sample and the scientific conclusion drawn from the study.

Version 2:

Reviewer comments:

Reviewer #1

(Remarks to the Author)

In Response 1 to Reviewer #1's question, the Figure R1 data is considered to be very important. However, this data cannot be found in the revised manuscript or the revised supplementary materials. If Figure R1 and the related discussion are added to the revised manuscript or supplementary, I would recommend the publication of this paper.

Reviewer #3

(Remarks to the Author)

The authors addressed all of my concerns. I recommend publishing the current version of the manuscript in Nature Communications.

Version 3:

Reviewer comments:

Reviewer #1

(Remarks to the Author)

I have received sufficient answers to all the questions I asked. I recommend that this paper be published.

Reply to Reviewers' Comments

Reviewer #1 (Remarks to the Author):

The authors present a detailed investigation of untwisted R-stacked WSe₂ bilayers and successfully identify the coexistence of AB and BA ferroelectric domains. The results are scientifically intriguing and contribute meaningfully to the field of twistrionics. However, I believe the current version of the manuscript requires further clarification and supporting data before it can be considered for publication. My concerns are outlined below:

Response 1:

We thank the reviewer for the positive assessment of our work and for the suggestions that have helped to improve our manuscript.

1. The reliability of the band alignment analysis critically depends on the quality of the R-stacked bilayer. While the authors mention the use of the “tear-and-stack” method, the manuscript lacks sufficient characterization data to confirm sample uniformity. Given the nature of the stacking process, the presence of bubbles or other inhomogeneities is likely. Therefore, comprehensive data such as AFM topography across the entire sample, PL/Raman mapping, and representative spectra should be provided to validate the sample quality.

Response 2:

We thank the reviewer for this important comment. We agree that the uniformity of the bilayer is critical for our analysis.

For this type of TMD homobilayer device, we argue that the quality of the buried interface is more critical than the topography of the outermost surface. In our device structure, the bilayer WSe₂ is fully encapsulated, buried beneath two layers of thick hBN with a combined thickness exceeding 100 nm. In this configuration, the information regarding the critical WSe₂/WSe₂ interface that can be reliably obtained from AFM topography is limited.

Therefore, to validate the sample quality, we provide comprehensive optical characterization, including bright-field (BF), dark-field (DF), and photoluminescence (PL) images. These data are now included in the Supplementary Information as Figure R1.

From the BF and DF images, the boundaries of the individual monolayers are clearly visible. Within the bilayer region of interest, only a few, very small bubbles can be observed.

More significantly, the PL mapping provides crucial insight into the interlayer coupling. As shown in Figure R1(c), the monolayer WSe₂ region exhibits strong and uniform PL, indicating the high intrinsic quality of the material. In sharp contrast, the PL in the entire bilayer region is almost completely quenched. This strong and highly uniform quenching is a clear signature of robust and homogeneous interlayer coupling across the stacked area.

Furthermore, we note that the few bubbles visible in the BF/DF images do not appear as local bright spots in the PL map. This strongly suggests that these bubbles are not located at the critical WSe₂/WSe₂ interface (which would disrupt the quenching), but are likely located at other interfaces, such as between the graphene back-gate and the bottom hBN layer.

These optical data collectively demonstrate the high uniformity and excellent interface quality of our bilayer WSe₂ sample. We have added these images and discussion to the Supplementary Information (new Figure S1) to validate the sample quality for all readers.

Figure R1: Optical characterization and schematic of the bilayer WSe₂ device. (a) Bright-field (BF) and (b) Dark-field (DF) optical images of the device. The boundaries of the monolayer (ML) and bilayer (BL) WSe₂ regions are clearly visible. (c) Photoluminescence (PL) intensity image of the sample. The ML region shows strong, uniform PL, whereas the BL region exhibits strong and highly uniform PL quenching, indicating robust and homogeneous interlayer coupling. (d) Device schematic illustrating the ML and BL areas, top and bottom gates, and bubbles. The experimental data were collected from the dual-gated BL region, which these optical data collectively demonstrate is highly uniform and possesses a high-quality interface.

2. As illustrated in Fig. 1b, domains induced by stacking are expected to exhibit symmetry or regular patterns. However, the KPFM data shown in Fig. S1 does not clearly demonstrate such regularity. The authors are encouraged to explain this discrepancy and discuss possible reasons for the observed domain distribution.

Response 3:

We thank the reviewer for this insightful question.

The reviewer is correct that homobilayers with a small twist angle typically exhibit periodic, regular reconstruction patterns. However, this behavior changes dramatically as the twist angle approaches exactly 0° and the interlayer strain is minimized. In this near-ideal alignment limit, the domain size is expected to increase significantly, and the domain structure becomes less periodic and more irregular. In the most idealized scenario of a perfectly aligned, strain-free bilayer, the entire sample would form a single domain. These large irregular domains are often observed in naturally 3R-stacked MoS₂ (e.g., *Nature* **612**, 465–469 (2022)).

In artificially stacked homostructures, some degree of residual strain or a minimal twist angle is often unavoidable, which typically results in the formation of periodic reconstruction patterns reported in many studies. Therefore, the observation of very large and irregular domains in our KPFM data (new Figure S2) is not a discrepancy, but rather strong evidence for the exceptionally high quality of our sample, indicating a near-perfect 0° twist angle and minimal strain.

We also thank the reviewer for pointing out that our original schematic in Figure 1b could be misleading. We have now revised Figure 1b to depict two larger, more distinct domains, which

more accurately reflects the physical reality of our high-quality, 0° twisted sample.

3. It is unclear which specific region of the sample was selected for the measurements presented in Fig. S1. Can identical optical properties be consistently observed across all domains that exhibit similar contrast in Fig. S1? If so, please provide supporting evidence. Furthermore, is the data reproducible even within the same domain? Clarification and corresponding data are requested. Since multiple domains are visible, a more detailed justification for the choice of measurement location would significantly enhance the credibility of the experimental results.

Response 4:

We thank the reviewer for raising these important points regarding reproducibility and measurement location. We apologize for the initial lack of clarity.

We must first clarify a critical experimental detail. The KPFM measurements (shown in new Figure S2) must be performed on a sample surface and thus were taken on a non-encapsulated bilayer. In contrast, the low-temperature optical measurements, which are central to our work, require the sample to be fully encapsulated in hBN to protect the material and enable electrostatic gating. Therefore, the optical measurements were necessarily performed on a separate, fully encapsulated device. It is not feasible to perform KPFM on the actual encapsulated device used for the optical experiments.

The purpose of new Figure S2 is to provide independent, supporting evidence that our R-stacked bilayer WSe₂ material inherently forms ferroelectric domains (AB vs. BA stacking) with distinct, measurable surface potentials. This is a known characteristic of this material system. To prevent this confusion for future readers, we have revised the caption for new Figure S2 to make the sample difference and the figure's purpose explicit. We have added two sentences to the caption "This measurement was performed on a separate sample from the main encapsulated (gated) device used for optical measurements, as KPFM requires an exposed surface. The purpose of this figure is to provide independent validation of intrinsic ferroelectric domain formation."

However, to thoroughly address the reviewer's valid concerns about uniformity and reproducibility on the device used for our main results, we have performed extensive 4K reflectance mapping experiments on the same device in the main text.

Figure R2(a) shows a 4K reflectance map of the device, generated by integrating the spectral intensity from 1.65 eV to 1.70 eV (a range influenced by both WSe₂ and graphite absorption). The map, which includes a schematic of the device structure, confirms that the optical properties across the entire bilayer (BL) WSe₂ area are highly uniform.

To further demonstrate this consistency, Figure R2(b) presents reflectance contrast spectra from nine different locations along the red and white star direction. While there are, as expected, minor variations in peak position, intensity, and full-width at half-maximum (FWHM), all nine spectra clearly exhibit the two distinct absorption peaks corresponding to the LE and HE excitons.

Furthermore, to explicitly confirm the reproducibility of the core physical phenomena, we performed complete doping-dependent reflectance measurements at four additional locations (presented in Figure R2(c-f)), separate from the data in the main text. Although the overall data quality varies between spots (for instance, the signal in Figure R2(e) is visibly weaker than in other locations), the spectral evolution as a function of doping is completely consistent across all measured locations. This robustly demonstrates that the exciton-polaron physics we report is a general and reproducible property of the device, not an artifact of a single, specially chosen

location.

The data presented in the main manuscript was collected from the locations marked by the red and white stars in Figure R2(a). We selected these specific points because they exhibit the narrowest and strongest reflectance peak, which allows the underlying physical phenomena to be presented with the greatest clarity for the reader. Furthermore, these spots are highly representative: the red star is in a region with multiple domains, while the white star is in a large single-domain region, confirming our conclusions hold across different local domain environments.

We have added these data the Supplementary Information (new Figure S3) to validate the data reproducibility for all readers.

Figure R2 | Optical uniformity and reproducibility of the encapsulated device. (a) 4K reflectance map of the device, generated by integrating the spectral intensity between 1.65 eV and 1.70 eV. A schematic of the device stack is overlaid for clarity. The reflectance signal is influenced by both the WSe₂ bilayer (BL) and the graphite gates. The map demonstrates the high spatial uniformity of the BL WSe₂ area. The red and white stars indicate the measurement locations for the data presented in the main manuscript, corresponding to a mixed-domain and a large single-domain region, respectively. (b) Reflectance contrast spectra from nine locations along the red and white star direction. Despite minor variations in peak characteristics (position, intensity, and width), all spectra clearly resolve the distinct LE and HE exciton peaks. (c-f) Doping-dependent reflectance contrast spectra collected from four additional locations on the device (distinct from those in the main text). While the signal-to-noise ratio varies (e.g., weaker intensity in (e)), the spectral evolution trend with doping is identical across all locations, confirming the reproducibility of the reported exciton-polaron physics.

4. The abstract mentions “establishing key parameters to understand twisted bilayers and enabling new ferroelectric and excitonic device opportunities,” and a similar statement appears in the conclusion. To strengthen the impact of the manuscript, the authors should elaborate on how their findings could be applied in practical device contexts, with specific examples or mechanisms.

Response 5:

We thank the reviewer for this valuable suggestion to strengthen the manuscript's impact. We agree that elaborating on the practical device applications of our findings is crucial. To address this point effectively, we believe the conclusion section is the most appropriate place for this detailed discussion and future outlook, as the abstract must remain a concise summary of our primary results.

Therefore, we have expanded the final paragraph of our conclusion to provide specific examples and mechanisms. The added text is as follows:

"Specifically, the electric-field control over coexisting ferroelectric domains, which we demonstrate, is a key mechanism for developing ultra-low power sliding ferroelectric memories. As recent work has highlighted, these systems show great promise for applications such as neuromorphic computing (e.g., as artificial synapses) [Li, Xiuzhen, et al. *Nature Communications* 15.1 (2024): 10921.] and sequential logic-in-memory circuits [Niu, Jingjie, et al. *ACS nano* 19.5 (2025): 5493-5502.]. Furthermore, coupling these controllable ferroelectric domains with other intrinsic 2D properties, such as the spin-valley locking in WSe₂, could enable novel spintronic devices like spin field-effect transistors or spin logic gates. This platform also offers a new route to electrically switch or tune quantum phases, such as superconductivity, in related moiré twisted bilayers."

We believe this addition directly addresses the reviewer's comment by connecting our fundamental findings to practical device concepts, thereby strengthening the manuscript's impact.

In summary, while the manuscript focuses on deep scientific insights, additional explanation and supporting data are necessary to improve its accessibility and impact. In particular, given the experimental complexity, the quality of the sample plays a crucial role. I strongly recommend the inclusion of more thorough sample characterization to ensure the reliability of the reported results.

Response 6:

We thank the reviewer again for the constructive comments on improving our manuscript.

Reviewer #2 (Remarks to the Author):

The manuscript entitled "Highly tunable band structure in ferroelectric R-stacked bilayer WSe₂" by Zhe Li et al. presents interesting results on the analysis of key parameters affecting the electronic band structure of untwisted R-stacked bilayer WSe₂ using optical measurements. The device scheme involving a dielectric-sandwiched R-stacked bilayer WSe₂ to understand doping via the same polarity gating and ferroelectric built-in potential via opposite polarity gating is a novel approach to understanding the complex type-II band alignment in R-stacked bilayer WSe₂. Their experimental results and the proposed model for understanding the band-gap difference, interlayer potential, and band offset in R-stacked bilayer WSe₂ are informative and convincing. Thus, I suggest minor revisions before this manuscript can be considered suitable for publication in *Nature Communications*.

Response 7:

We thank the reviewer for appreciating our experimental results and model.

i) The authors present KPFM data in Fig. S1 to confirm that the device stacking is R with either BA or AB stacking order in bilayer WSe₂. However, the surface potential values for both stackings are positive. Could the authors explain why both stackings show positive values,

given that the built-in potential for BA and AB R-stacked WSe2 should be opposite? Also, is KPFM data alone sufficient to prove that the WSe2 bilayer is untwisted R-stacked?

Response 8:

We thank the reviewer for the valuable comments. The reviewer correctly notes that the built-in potential for AB and BA R-stacked domains should theoretically have opposite signs. The positive potentials observed in our new Figure S2 are due to a significant systematic offset inherent in the KPFM measurement. KPFM accurately measures the relative potential difference between the domains, which is sufficient to confirm the existence of two distinct electronic states (AB and BA). However, the absolute potential value is highly susceptible to external factors such as stray electric fields from the substrate, ambient contaminants, and the precise condition of the tip. In our case, these factors introduced a large positive offset, shifting the entire potential map. We chose to present the raw, measured data rather than apply an arbitrary zero-point shift to force the values to be opposite. We have also included this discussion into new Figure S2.

Regarding the second point, the reviewer is correct that KPFM data alone is insufficient to definitively prove the WSe2 bilayer is R-stacked. Our conclusion is based on the combination of KPFM data and our fabrication method. The KPFM images, by showing large-scale domains (micrometers) rather than a small-period moiré pattern, strongly indicate that the twist angle between the layers is very close to 0°.

This near-zero twist angle is a direct consequence of our "tear-and-stack" fabrication technique [Kim, K. et al. *Nano Letters* 16, 1989–1995 (2016)]. This method involves tearing a single WSe2 monolayer flake and then precisely stacking one piece back onto the other. Because both layers originate from the exact same parent crystal, their relative orientation can be controlled with high precision. By stacking them without intentional rotation, we ensure their crystal axes are aligned (parallel), which defines the R-stacking configuration (near 0° twist angle). Therefore, the KPFM data and our fabrication method confirms the untwisted R-stacking.

ii) Given the broad readership of Nature Communications, the authors should provide more detailed information on key experimental parameters and the calculation process. Specifically, the manuscript should present detailed information on the reflectance contrast ($\Delta R/R_0$ and normalized $\Delta R/R_0$). Please also include the calculation process for evaluating the doping concentration in R-stacked bilayer WSe2 using dual gating, as well as the calculation for the g-factor. Additionally, the g-factor for AP- is slightly different from that of the neutral excitons. Can these values be treated as identical, as mentioned on page 3?

Response 9:

We thank the reviewer for these constructive suggestions, which help improve the clarity and completeness of the manuscript for a broad readership.

We have addressed all points by adding a detailed methods section to the Supplementary Information (Section S1). This new section provides:

1. **Reflectance Contrast ($\Delta R/R_0$):** A detailed explanation of how the differential reflectance contrast $\Delta R/R_0 = (R_{\text{sample}} - R_0)/R_0$ is calculated, where R_0 is the background reflectance from a substrate area without the WSe2 bilayer.
2. **Normalized Reflectance Contrast:** A clarification of the normalization procedure used in Figure 3c. We explain that the integrated intensity of the X_H^0 (which serves as a proxy for oscillator strength) is normalized to its maximum value. This normalization allows for the direct comparison of the relative contributions from the coexisting AB

and BA domains.

3. **Doping Concentration Calculation:** A step-by-step description of the calculation used to evaluate the doping concentration n induced by the dual-gate structure, based on a parallel-plate capacitor model.
4. **g-factor Calculation:** The full procedure for extracting the g-factors. We show the helicity-resolved spectra in the new Figure S5 and detail the fitting process for the Zeeman splitting, which is linearly fitted to extract g (new Figure S6).

Regarding the second point on the AP⁻ g-factor: The reviewer is correct that its g-factor is slightly different from that of the neutral excitons, and our original text was imprecise. We have revised the main text (page 3) to state that the values are "similar" rather than "identical".

The g-factors of WSe₂ excitons are known to be sensitive to their dielectric environment [Liu, Erfu, et al. *Physical Review B* 99.20 (2019): 205420. Förste, Jonathan, et al. *Nature Communications* 11.1 (2020): 4539. Koperski, Maciej, et al. *2D Materials* 6.1 (2018): 015001. Robert, Cedric, et al. *Physical Review Letters* 126.6 (2021): 067403.]. Since AP⁻ is an attractive polaron (an exciton dressed by a Fermi sea), its g-factor is expected to be further modified by the doping level. This small discrepancy does not invalidate our main conclusion that in the AP⁻ case, the electrons are not doped into the K valley.

We believe these additions significantly strengthen the manuscript by providing the necessary experimental and computational details.

iii) On page 6, the authors suggest a method for measuring the internal potential in R-stacked WSe₂ by sweeping the external electric field D . How can the internal potential be determined by finding the external electric field D value that results in an equal energy difference between the attractive polarons and the neutral excitons of both layers?

Response 10:

We thank the reviewer for pointing out the lack of clarity in our description. We apologize for the confusion and have revised this section in the manuscript (Page 6) to provide a more detailed explanation based on the underlying physical mechanism.

Our method is based on tuning the spatial distribution of the doped Λ valley electrons using the external displacement field, D . The key physics is as follows:

At finite electron doping, the neutral excitons (X_L^0 and X_H^0) interact with the Λ valley electron gas. This many-body interaction (e.g., screening) renormalizes the exciton energy, forming the attractive polaron (AP) states. This process manifests as an energy redshift ($E_{\text{shift}} = E_X - E_{\text{AP}}$). The magnitude of this redshift E_{shift} is directly related to the strength of the exciton-electron interaction, which in turn depends on the spatial overlap between the exciton wavefunction (localized in the HE or LE layer) and the Λ valley electron gas wavefunction.

Due to the intrinsic $E_{\text{built-in}}$ in R-stacked WSe₂, the Λ valley electrons are pulled asymmetrically, locating them closer to the HE layer. This results in a stronger interaction and thus a larger energy redshift for the HE polaron compared to the LE polaron ($E_{\text{shift, H}} > E_{\text{shift, L}}$).

By applying an external field D , we can counteract the internal field and "push" the Λ electron gas. The internal potential is determined by finding the exact external field D that precisely cancels $E_{\text{built-in}}$. At this specific cancellation point ($D = E_{\text{built-in}}$), the Λ electron wavefunction becomes symmetrically distributed between the two layers. At this unique symmetric point, the electron gas interacts identically with both the HE and LE excitons. Consequently, their energy redshifts become equal: $E_{\text{shift, H}} = E_{\text{shift, L}}$.

Therefore, if we look at the polaron energy splitting,

$$\begin{aligned}\Delta E_{\text{AP}} &= E(\text{AP}_{\text{H}}^-) - E(\text{AP}_{\text{L}}^-) = [E(\text{X}_{\text{H}}^0) - E_{\text{shift, H}}] - [E(\text{X}_{\text{L}}^0) - E_{\text{shift, L}}] \\ &= \Delta E_{\text{X}^0} - [E_{\text{shift, H}} - E_{\text{shift, L}}].\end{aligned}$$

When we reach the symmetric point, $E_{\text{shift, H}} = E_{\text{shift, L}}$, the second term becomes zero. The equation simplifies to $\Delta E_{\text{AP}} = \Delta E_{\text{X}^0}$. Thus, the D field at which the polaron energy splitting equals the neutral exciton splitting is, by definition, the value of the internal $E_{\text{built-in}}$.

We have revised the main text on Page 6 to clarify this mechanism:

"We apply a constant electron doping concentration of $n = 2.0 \times 10^{12} \text{ cm}^{-2}$ and sweep the external displacement field D. The key insight is that when the external field exactly cancels the internal field, the Λ -valley electrons are symmetrically distributed between the two layers. At this symmetric point, the Λ electron gas interacts identically with both the HE and LE excitons, leading to an equal energy redshift for both polaron states. Consequently, the energy difference between the attractive polarons, AP_{L}^- and AP_{H}^- , becomes equal to the energy difference of the neutral excitons, X_{L}^0 and X_{H}^0 ."

iv) Regarding ferroelectric switching in R-stacked bilayer WSe2, why does only the p-doped R-stacked bilayer WSe2 show switching behavior in contrast to the n-doped R-stacked bilayer WSe2?

Response 11:

We thank the reviewer for this question, which allows us to clarify a key point. We apologize if our manuscript gave the impression that this behavior was unique to p-doping. In fact, we observe robust ferroelectric switching in n-doping as well.

To demonstrate this explicitly, we have performed additional D field dependent experiments with n-doping as shown in Figure R3.

- **Figure R3a** shows the **forward sweep**, where D is scanned from -0.12 V/nm to +0.12 V/nm (indicated by the red arrow). As the field becomes positive and its magnitude increases, we observe two distinct, abrupt jumps in the attractive polaron energies.
- **Figure R3b** shows the **backward sweep**, from -0.12 V/nm to -0.12 V/nm. Similarly, as the field becomes negative and its magnitude increases, we again observe two sharp discontinuities.

These abrupt jumps are the clear signature of ferroelectric domain switching. The fact that we observe two jumps in each sweep direction, rather than a single one, likely indicates that the domain boundary moves across our optical spot in two discrete steps.

These results unequivocally confirm that ferroelectric switching occurs in n-doped samples. This suggests that the type of charge carrier (electrons versus holes) does not significantly impact the fundamental mechanism of domain switching in this system. We have added this figure and a corresponding discussion to the Supplementary Information (new Figure S9).

Figure R3 | Ferroelectric switching in n-doped R-stacked bilayer WSe2. (a, b) Reflectance spectrum as a function of the displacement field D . The energies of the attractive polarons (AP) are tracked. (a) The forward sweep of the D field, from -0.12 V/nm to $+0.12$ V/nm. (b) The backward sweep of the D field, from $+0.12$ V/nm to -0.12 V/nm. The sweep direction is indicated by the red arrows. In both panels, two abrupt, discontinuous jumps in the polaron energies are observed, which are the characteristic signatures of ferroelectric domain switching. The presence of two jumps in each direction suggests a multi-step process for the domain wall to traverse the optical spot. This data confirms that robust switching is also present with n-doping.

v) In Fig. 1(b), the device schematic for untwisted R-stacked WSe2 with two domains could mislead readers since it looks like a twisted R-stacked WSe2 with a moiré structure. Please clarify that the orange or white domains represent opposite stacking orders, not a moiré structure.

Response 12:

We thank the reviewer for this valuable comment. We agree that our previous schematic in Figure 1(b) could potentially be misleading. As the reviewer correctly points out, the different domains (previously depicted as periodic orange and white regions) are intended to represent the two coexisting, opposite stacking orders (e.g., AB and BA stacking) in our high-quality, 0° twisted sample, and not a moiré structure.

To address this ambiguity and make the schematic clearer, we have revised Figure 1(b) in the new manuscript. The updated figure now depicts two larger, more distinct domains, clearly illustrating the presence of both AB and BA stacking configurations. We believe this revised representation more accurately reflects the physical reality of our device while avoiding any potential confusion with moiré patterns.

vi) Physical quantities should be italicized. For example, in Fig. 1(e), V should be italicized. Please correct any typos throughout the manuscript.

Response 13:

We thank the reviewer for the comments. We have now corrected typos throughout the manuscript.

vii) While I agree with the authors' claim that untwisted R-stacked bilayer WSe2 exhibits LE

and HE type-II band structures regardless of stacking order, it seems that optical measurements cannot determine which layer is LE or HE. Is there an efficient method to distinguish the LE layer from the HE layer in untwisted R-stacked bilayer WSe₂?

Response 14:

We thank the reviewer for this insightful question.

The reviewer is correct that for intralayer optical measurements, such as our reflectance spectroscopy, it is challenging to assign the LE and HE signals to the specific physical (e.g., top or bottom) layer.

However, an efficient optical method to make this distinction does exist by measuring the Stark shift of the interlayer exciton (IX) photoluminescence (PL) (as shown in new Figure S11).

Based on the established type-II band structure of R-stacked bilayer WSe₂, the conduction band minimum is at the Λ valley (spatially located between the two layers), while the valence band maximum is at the K valley of the LE layer. Therefore, the lowest-energy IX transition consists of an electron in the Λ valley and a hole physically localized within the LE layer.

This inherent charge separation of the IX allows us to use an external electric field (D) to pinpoint the LE layer's physical location. If we apply a D field pointing, for example, from the top layer to the bottom layer:

1. If the top layer is the LE layer: The hole is in the top layer. The applied field will increase the hole's potential energy (relative to the electron in the middle). This will cause the IX energy to increase (a blueshift).
2. If the bottom layer is the LE layer: The hole is in the bottom layer. The applied field will decrease the hole's potential energy. This will cause the IX energy to decrease (a redshift).

Thus, by observing the sign of the IX Stark shift (a blueshift or redshift) in response to a known field direction, one can unambiguously determine whether the top or bottom physical layer is the LE layer.

As shown in new Figure S11, we clearly observe these linear Stark shifts. The sign of this shift unambiguously determines whether the top or bottom physical layer is the LE layer within the measured ferroelectric domain. We have added the following discussion into the caption of new Figure S11:

"Furthermore, this field-dependent measurement provides an unambiguous method to identify the physical location of the LE and HE layers. The IX consists of an electron in the Λ valley (spatially located between the layers) and a hole localized in the K valley of the LE layer. A redshift (blueshift) under positive D indicates the hole is in the bottom (top) layer, meaning the bottom (top) layer is the LE layer."

Reviewer #3 (Remarks to the Author):

The manuscript entitled "Highly tunable band structure in ferroelectric R-stacked bilayer WSe₂" by Li et al. reveals the exciton complex in the R-stacked bilayer WSe₂. The data quality is high standard. The switching between the AB and BA stacks was detected in the photoluminescence spectra. Nevertheless, there are several concerns that the authors should address before it can be recommended for publication in Nature Communications.

Response 15:

We thank the reviewer for appreciating our data quality.

1. The size of the AB and BA domains is several micrometers. It indicates that the authors can investigate in AB and BA domains. The data shown in Figs. S4 and S5 were claimed in a single domain. Why did the author choose the data from mixed domains to analyze in the main manuscript? This makes the data analysis more complicated, and it is difficult for the reader to follow.

Response 16:

We thank the reviewer for the comments and this important question regarding our data selection.

We chose to present the data from the mixed-domain region (Figure 3) in the main manuscript precisely because it offers a powerful, side-by-side demonstration of ferroelectricity. This dataset uniquely illustrates the simultaneous and opposite response of adjacent AB and BA domains to the same external electric field. This is the direct hallmark of their inverted band structures and spontaneous polarization, providing compelling evidence in a single measurement.

Furthermore, we believe this data is more representative and of greater value to the wider research community. While our high-quality sample exhibits large, micrometer-scale domains (as shown in KPFM in new Figure S2) that allow us to isolate single domains (data in new Figure S7, S8), this is not the typical case. In many R-stacked samples, the domain sizes are on the order of several hundred nanometers. For any far-field optical measurement (which typically uses a laser spot of $\sim 1 \mu\text{m}$), probing such a sample will inevitably result in averaging over a mixed-domain region.

Therefore, our analysis of the mixed-domain data in the main text provides a crucial framework for other researchers to interpret their own optical data, which will most likely originate from mixed-domain areas. We have now added a sentence to the main text to clarify this rationale:

"Based on the relative intensity change, we can estimate that the laser spot covers an area with an AB to BA domain ratio of approximately 1:1 for this particular sample position. See new Figure S7 and S8 for data at another spatial position which has only a single ferroelectric domain. We note that the data from mixed-domain region not only provides a powerful, simultaneous demonstration of the opposite AB/BA response but is also more representative of typical R-stacked samples where domain sizes are often smaller than the optical spot."

2. Compare the data of Fig. 3a (multidomain), Fig. S4b (single domain), and Fig. S5 (single domain); it is difficult to see the difference. Could the author provide more analysis to make the claim of multidomain more convincing? Could the authors provide the spectra at that specific multidomain and compare them with the corresponding spectra of the single domain?

Response 17:

We thank the reviewer for the comments.

To make the distinction between the multi-domain and single-domain regions more clear, we have created a new comparison figure (presented as Figure R4 in this response) that directly compares the reflectance maps at the same fixed field of $D = -0.025 \text{ V/nm}$.

- Figure R4a shows the multi-domain region (from main text Figure 3a).

- Figure R4b shows the Type 1 single domain (from new Figure S7).
- Figure R4c shows the Type 2 single domain (from new Figure S8).

To help the reader quickly focus on the critical area of interest, we have added red dashed boxes to highlight the high hole-doping regime in all three panels.

By comparing these highlighted regions, the difference is now stark:

1. In the single domain (b), the red box shows that the neutral exciton (X_H^0) retains its oscillator strength. This means the external field pushed the HE layer valence band down, making it harder to populate with holes, so no polarons form.
2. In the other single domain (c), the red box shows that the X_H^0 oscillator strength has vanished, and is completely replaced by the attractive and repulsive polarons. This is because the external field pushed the HE layer valence band up, making it easier to populate.
3. In the multi-domain region (a), the red box clearly shows a superposition of both behaviors: the X_H^0 peak persists (like domain b) with a reduced oscillator strength, but at the same time, the attractive and repulsive polaron branches also clearly appear (like domain c).

This simultaneous observation of two opposite responses is the definitive proof that the area in Figure 3a contains both AB and BA domains. To ensure this is clear for all readers, we have also added these red dashed boxes to the original figures in the Supplementary Information (new Figure S7 and S8).

Figure R4 | Direct comparison of multi-domain and single-domain responses at a fixed electric field. Doping-dependent reflectance maps for (a) the multi-domain region, (b) a single domain (Type 1), and (c) the other single domain (Type 2). The red dashed boxes highlight the high hole-doping regime. The single domains show a clear, opposite response: in (b), the X_H^0 peak persists (doping suppressed), while in (c), it is replaced by polarons (doping enhanced). The multi-domain region in (a) clearly shows a superposition of both signatures, confirming the coexistence of AB and BA domains.

3. Could the authors provide a similar analysis of Fig. 3c for Fig. S4 and Fig. S5?

Response 18:

We thank the reviewer for the comments.

As requested, we have performed the same analysis as in Figure 3c (plotting the X_H^0 oscillator strength versus doping) for the single-domain data taken at the same electric field of $D = -0.025$ V/nm (data from new Figure S7b and Figure S8b).

The results are now plotted in Figure R5. The two curves show a starkly different behavior:

- For the first single domain (from Figure S7b), the X_H^0 oscillator strength decreases slowly and gradually as the hole concentration increases. This minor decrease is likely due to the changing dielectric environment as the entire sample is doped. Crucially, even at a high doping of $n \approx -10.0 \times 10^{12} \text{ cm}^{-2}$, the X_H^0 strength remains at a high level, confirming that holes are not doping into the HE layer in this domain.
- In sharp contrast, for the second single domain (from Figure S8b), the X_H^0 strength drops rapidly at a threshold of $n \approx -7.0 \times 10^{12} \text{ cm}^{-2}$, quickly falling to a low level. This is the clear signature of holes doping into the HE layer, causing the X_H^0 exciton to convert into attractive and repulsive polarons.

This direct, quantitative comparison of the two single domains provides definitive proof of their opposite response to the applied electric field. It further validates our analysis of the multi-domain region (Figure 3c), which showed a partial, plateaued drop in intensity consistent with a superposition of these two behaviors.

Figure R5 | Quantitative comparison of X_H^0 oscillator strength in single domains. Oscillator strength of the high-energy neutral exciton (X_H^0), plotted as a function of hole doping concentration (n). The data is extracted from the single-domain maps in new Figure S7b and S8b, both at $D = -0.025$ V/nm.

4. In Figure 3b, the appearance of the Q valley is the lowest energy point. Is the indirect exciton from the K-K valley in Fig. 3e correct? Is it possible from the Q-K valley?

Response 19:

We thank the reviewer for this very sharp observation.

As shown in our band diagram (Figure 3b), the Q(Λ) valley is indeed the conduction band minimum (CBM). Consequently, the lowest-energy interlayer exciton (IX) is the Q-K transition, which involves an electron at the Q-valley and a hole at the K-valley.

This is precisely what we observe in our photoluminescence (PL) spectra in new Figure S11. As shown there, the Q-K exciton emission appears at a much lower energy (around 1.5 eV at

$D = 0$) than the intralayer excitons.

However, the data presented in Figure 3d is from reflectance spectroscopy, which differs fundamentally from PL. While PL spectroscopy is dominated by emission from the lowest-energy relaxed state (the Q-K exciton) due to efficient carrier relaxation, reflectance spectroscopy can probe other, higher-energy excitonic states with larger oscillator strength.

The IX feature observed in Figure 3d (which anti-crosses with the intralayer excitons) has an energy that is very close to the intralayer transitions. Given that the Q-K exciton's energy is significantly lower (~ 1.5 eV), the transition we observe in the reflectance spectrum in Figure 3d cannot be the Q-K exciton. Therefore, we identify it as a well-known, higher-energy interlayer exciton: the K-K transition.

5. Could the author provide the data of AB and BA PL, as shown in Fig. 1c? It is interesting to see how it is different.

Response 20:

We thank the reviewer for the comments. We have now extracted multiple PL spectra from new Figure S11 and plotted as Figure R6.

- **Figure R6a** shows the PL spectra at zero displacement field ($D=0$). As can be seen, the two spectra are nearly identical. This is because although the AB and BA domains possess opposite built-in electric fields, their electronic states are degenerate at $D=0$. Their respective interlayer excitons (IX) have oppositely oriented dipoles, but their energies are the same, resulting in overlapping emission peaks.
- **Figure R6b** shows the spectra under an applied external field of $D = 0.03$ V/nm. Here, the difference is clear. The external field breaks the degeneracy. Because the intrinsic IX dipoles in the two domains point in opposite directions, the applied field induces an opposite Stark shift: one peak shifts to a lower energy (redshift) while the other shifts to a higher energy (blueshift).

This new PL data provides further direct confirmation of the opposite ferroelectric nature of the AB and BA domains.

Figure R6 | Photoluminescence (PL) spectra of AB and BA domains. (a) At zero displacement field ($D = 0$). The interlayer exciton (IX) emission peaks from both domains are degenerate, resulting in nearly identical spectra. (b) Under an applied displacement field of $D = 0.03$ V/nm. The external field lifts the degeneracy. Due to the oppositely oriented dipoles of the interlayer excitons in the AB and BA domains, their emission peaks exhibit an opposite Stark shift (one blueshift and one redshift), confirming their distinct ferroelectric nature.

6. Could the author provide the data for the positive D field, as in Fig. 4a for the negative D field?

Response 21:

We thank the reviewer for this suggestion. We have performed this measurement (Figure R7) and included the data in the Supplementary Information (now Figure S9).

This figure shows a full sweep of the displacement field D from -0.12 V/nm to $+0.12$ V/nm (and back). This dataset provides the data for the positive D field range as requested. It also clearly captures the ferroelectric domain switching in the n-doped sample, which is visible as abrupt jumps in the polaron energies. This figure is now available in the Supplementary Information.

Figure R7 | Ferroelectric switching in n-doped R-stacked bilayer WSe2. (a, b) Reflectance spectrum as a function of the displacement field D . The energies of the attractive polarons (AP) are tracked. (a) The forward sweep of the D field, from -0.12 V/nm to $+0.12$ V/nm. (b) The backward sweep of the D field, from $+0.12$ V/nm to -0.12 V/nm. The sweep direction is indicated by the red arrows. In both panels, two abrupt, discontinuous jumps in the polaron energies are observed, which are the characteristic signatures of ferroelectric domain switching. The presence of two jumps in each direction suggests a multi-step process for the domain wall to traverse the optical spot. This data confirms that robust switching is also present with n-doping.

In addition, I have other comments:

- The introduction part is not convincing to me. The relaxation at the micrometer AB and BA deteriorates the flat band. The citation in this part is for STM, with AB and BA at the nanometer scale. That part of the introduction should be revised.

Response 22:

We thank the reviewer for the comments. The reviewer correctly notes that the moiré flat bands discussed in Refs. [18, 19] arise from the periodic superlattice formed by nanometer-scale AB and BA domains, which occurs in small-angle twisted bilayers. In contrast, our work focuses on the 0-degree (untwisted) limit. In this regime, the lattice fully relaxes into large, micrometer-scale domains. As the reviewer rightly implies, this configuration does not host moiré flat bands,

as it lacks the periodic moiré potential.

Our intention in citing Refs. [18, 19] was to establish the critical importance of the R-stacked ferroelectric domain as the fundamental "building block" whose properties are essential in both physical regimes:

1. In the twisted case, the periodic potential landscape created by these nm-scale ferroelectric blocks is what generates the flat bands.
2. In our untwisted case, the large micrometer-scale of the domains allows us to isolate and study the intrinsic physical properties of this "building block" itself (e.g., its built-in field and band alignment) directly, without the complexities of a moiré superlattice.

We have revised this paragraph in the introduction (Page 1) to make this distinction clear. The following sentence in the introduction have been changed from:

"A crucial feature of tWSe2 at near-zero twist angles is that the lattice reconstructs into extended R-stacked ferroelectric domains (see schematic in Fig. 1b), providing the building blocks for moiré superlattices with flat bands and correlated states [18, 19]. "

To:

"A crucial feature of tWSe2 at near-zero twist angles is that the lattice reconstructs into a periodic superlattice of nanometer-scale ferroelectric domains, providing the building blocks for moiré superlattices with flat bands and correlated states [18, 19]. In the 0-degree (untwisted) limit, which is the focus of our study, these domains further relax into large, micrometer-scale regions (see schematic in Fig. 1b). "

- Fig. 1a: Adding a top view of the structure will be useful.

Response 23:

We thank the reviewer for the comments. A top view is added in Figure 1a.

- Figure 1d: Exaggerate bandgap differences, make the figure more informative.

Response 24:

We thank the reviewer for this suggestion. We have now exaggerated bandgap differences in Figure 1d to make the figure more informative.

- Single spectrum (e.g., Fig. 1c) with different regions?

Response 25:

We thank the reviewer for the comments.

We have performed extensive spatial-dependent measurements to validate this, as shown in Figure R8.

1. **Our sample is highly uniform.** As shown in the 4K reflectance map (Figure R8a), the optical properties across the entire bilayer WSe2 area are consistent.
2. **Spectra from different regions are nearly identical.** We collected reflectance spectra from nine different spot along the red and white star direction. (Figure R8b). All nine spectra are exceptionally similar, clearly resolving the same Low-Energy (LE) and High-Energy (HE) exciton peaks.

This high degree of similarity is expected, even when probing different ferroelectric domains (AB vs. BA) at zero external field ($D = 0$). There are two key physical reasons for this:

First, as we demonstrate with PL measurements in **Response 19**, the electronic states of the AB and BA domains are degenerate at $D = 0$. Although they possess opposite built-in fields, their energy levels are identical, leading to overlapping spectral peaks.

Second, the main features in our reflectance spectrum (Figure 1c) are the intralayer excitons. These excitons are charge-neutral, with their electron and hole wavefunctions confined within the same physical layer. This makes them, by nature, highly insensitive to electric fields (i.e., they exhibit a negligible Stark shift). Therefore, the opposite internal built-in fields of the AB and BA domains do not cause a measurable energy shift in these intralayer reflectance peaks.

In summary, the spectrum in Figure 1c is highly representative of the entire device.

Figure R8 | Optical uniformity and reproducibility of the encapsulated device. (a) 4K reflectance map of the device, generated by integrating the spectral intensity between 1.65 eV and 1.70 eV. A schematic of the device stack is overlaid for clarity. The reflectance signal is influenced by both the WSe₂ bilayer (BL) and the graphite gates. The map demonstrates the high spatial uniformity of the BL WSe₂ area. (b) Reflectance contrast spectra from nine locations along the red and white star direction. Despite minor variations in peak characteristics (position, intensity, and width), all spectra clearly resolve the distinct LE and HE exciton peaks.

Reply to Reviewers' Comments

Reviewer #1 (Remarks to the Author):

Most of my questions have been addressed with fairly thorough answers. However, I have found a serious problem in the interpretation of the experimental results. Specifically, the two different domains, which are the core of this paper, must be clearly distinguished. The domains were clearly identified in Figure S2 using KPFM, and that part is relatively clear. However, the optical analysis was conducted on an entirely different sample, not the one confirmed by KPFM. Figure R8 presents such a sample image.

The problem lies in the method used to distinguish the two different domains. The authors attempted to differentiate the domains by using reflectance data, marking the positions of the two domains with white and red stars. Yet, based on the presented results, it is difficult to objectively distinguish the domains. In other words, there is no direct evidence that the positions marked by the red and white stars correspond to different domains.

Figure R8(b) shows changes in the reflectance contrast spectra, but these signals can also vary depending on the sample quality (e.g., interface contaminations, bubbles, wrinkles). In fact, Figure S1 BF/DF images reveal that many bubbles are present in the sample. Figure S1(d) marks the positions of bubbles, and the authors argue in Figure S3 that measurements were taken at clean locations. However, microscopic bubbles or contamination cannot be identified solely with an optical microscope. This means that there could be subtle impurities at the positions marked by the red and white stars that are not distinguishable optically.

Therefore, more definitive experimental evidence is required. Unless direct proof is provided that the key results discussed in the main text were indeed measured from different domains, the experimental results cannot be considered reliable at all.

Response 1:

We thank the reviewer for this critical observation. We understand the concern regarding the objective identification of domains in the optical measurement sample, especially given the presence of bubbles. To address this, we have compiled a comprehensive rebuttal combining 1) **new experimental data (KPFM on a control sample)** confirming that bubbles in a few specific locations do not impede the formation of ferroelectric domains, 2) **comparison with state-of-the-art literature**, and 3) **detailed spectroscopic analysis** proving the effects of the domains is distinct from artifacts.

1. Integrity of the WSe2 Interface and New Control Experiment (Figure R1)

Regarding the bubbles: Our device consists of 8 layers (Topmost hBN, Top Graphite Gate, Top hBN, Graphite Contact, WSe2 Layer 1, WSe2 Layer 2, Bottom hBN, Bottom Graphite Gate). The critical interface is the WSe2/WSe2 interface, fabricated using the mature "tear-and-stack" technique, widely proven to produce high-quality homostructures [1-3]. The bubbles observed in optical images are statistically highly likely to be located at non-critical interfaces (e.g., between graphite and hBN) rather than the twisted WSe2 interface.

To verify this experimentally, we fabricated a new control sample (Figure R1) consisting of 2 nm top hBN / WSe2 / WSe2 / Graphite contact / Bottom hBN / Bottom Graphite gate.

- Optical Inspection: As shown in **Figure R1a** and **R1b**, bubbles are clearly visible in this sample, with a density and size distribution similar to the device used in the main text.
- KPFM Measurement: Despite the visible bubbles, KPFM measurements on this sample

reveal clear, well-defined ferroelectric domains (**Figure R1c**).

Figure R1 | Verification of ferroelectric domain formation in a control sample. (a) Bright and (b) dark microscope image of the control heterostructure (2 nm top hBN / R-bilayer WSe₂ / graphite contact / bottom hBN / bottom graphite gate). Similar to the device in the main text, bubbles are clearly visible across the sample. (c) Kelvin Probe Force Microscopy (KPFM) surface potential map of the red dashed boxes in (a) and (b). Despite the presence of bubbles visible in optical image, the KPFM scan reveals clear, well-defined ferroelectric domains with distinct potential contrast. This confirms that the bubbles generally reside at non-critical interfaces (e.g., hBN/graphite) and do not disrupt the formation of the ferroelectric moiré interface created by the tear-and-stack technique.

This result is crucial: it demonstrates that the presence of optically visible bubbles does not prevent the formation of ferroelectric domains at the WSe₂ interface. The "tear-and-stack" method remains robust against bubbles in the encapsulation layers.

2. Comparison with State-of-the-Art Literature (Figure R2)

We further note that bubbles are often inevitable in complex, multi-layer van der Waals heterostructures. Recent high impact works, for instance ref [2] and [4], utilizing similar multi-layer stacks also exhibit comparable density of bubbles (often less visible due to specific hBN thickness interference and lack of dark-field images that highlight bubbles more effectively). The community consensus is that while bubbles exist, they do not invalidate the physics observed in the clean regions of the device.

[Figure Redacted]

Figure R2 | Bubbles in state-of-the-art van der Waals heterostructures. Optical micrographs of high-performance devices reported in recent literature: (a) [2] and (b) [4]. These studies represent the current state-of-the-art in 2D material research. Note that despite the high quality of the physical results reported, bubbles (indicated by yellow arrows) are inevitably formed during the assembly of complex, multi-layer stacks. This comparison indicates that the presence of bubbles in the encapsulation layers does not preclude the existence of pristine interfaces and reliable physics in the active regions of the device.

In addition to visual inspection, we emphasize that in spectroscopic studies of 2D materials, the linewidth of the reflection spectrum serves as a more rigorous and quantitative standard for assessing sample quality. Our devices exhibit exceptionally narrow linewidths: the Full Width at Half Maximum (FWHM) of the reflection spectra reaches 3.6 meV for the monolayer, and 7.2 meV and 9.8 meV for the bilayer. These values match the highest standards reported in the literature for corresponding materials [5,6], providing solid spectroscopic evidence of the high crystalline quality and pristine interfaces of our samples, independent of the localized bubbles.

3. Spectroscopic Evidence: Distinct Domain Responses (Figure R3)

The reviewer asks for direct evidence that the red and white stars correspond to different domains rather than local impurities. We respectfully submit that the reflectance spectra themselves provide this definitive proof – a key finding of our manuscript. Due to the specific stacking orders (AB vs. BA), the two domains must exhibit opposite responses to the same external electric field.

Figure R3 | Direct comparison of multi-domain and single-domain responses at a fixed electric field. (a) single domain Type 1, (b) single domain Type 2 and (c) multi-domain region. The red dashed boxes highlight the high hole-doping regime. The simultaneous observation of two opposite responses in (c) serves as definitive proof that this area contains both AB and BA domains. (d) Schematic showing the opposite response of AB and BA domains to an external electric field. (e) Normalized peak intensity of the high-energy neutral exciton, plotted as a function of hole doping concentration (n).

In **Figure R3** (above), we directly compare the doping-dependent reflectance maps of the multi-domain region against single-domain regions. To facilitate the comparison, we have added red dashed boxes to highlight the high hole-doping regime.

- **Single Domain (Type 1, Panel a):** In the red box, the neutral exciton (X_H^0) retains its oscillator strength. This indicates the external field pushes the high energy layer valence band down, suppressing hole population (no polarons form). The schematic of the band structure change and doping-dependent intensity change are shown in **Fig.**

R3d and **R3e**, respectively.

- **Single Domain (Type 2, Panel b):** In the red box, the X_H^0 oscillator strength vanishes and is replaced by attractive and repulsive polarons. Here, the field pushes the valence band up, facilitating hole population.
- **Multi-domain Region (Panel c):** The red box clearly shows a superposition of both behaviors. The X_H^0 peak persists (characteristic of Type 1) but with reduced intensity, while the polaron branches (characteristic of Type 2) appear simultaneously.

This simultaneous observation of two opposite physical responses is the "fingerprint" of a multi-domain spot. This clear fingerprint is enabled by very clean optical spectroscopy of a R-WSe₂ with a pristine (ferroelectric) interface. This effect cannot be replicated if bubbles or contamination is present at the interface: spectroscopically, this would typically broaden lineshapes or reduce overall intensity. Rather, two distinct doping-dependent behaviors (Fig R3a and R3b) are superimposed in the mixed domain situation. We have added these clarifications and the marked figures to the Supplementary Information as new **Figure S9**.

In summary, the specific "superposition" optical signal combined with the new KPFM control data confirms that our results arise from ferroelectric domains, not sample contamination.

References:

- [1] Y. Cao *et al.*, Nature **556**, 43 (2018).
- [2] J. Cai *et al.*, Nature **622**, 63 (2023).
- [3] Y. Xia *et al.*, Nature **637**, 833 (2025).
- [4] R. Qi *et al.*, Science **390**, 299 (2025).
- [5] F. Cadiz *et al.*, Phys. Rev. X **7**, 021026 (2017).
- [6] KM McCreary, *et al.*, Nanoscale **14**, 147 (2022).

Reviewer #2 (Remarks to the Author):

The revised manuscript entitled "Highly tunable band structure in ferroelectric R-stacked bilayer WSe₂" by Zhe Li *et al.* have satisfactorily addressed all my previous comments and made corresponding changes to the manuscript and supplementary information. Now I recommend the study "Highly tunable band structure in ferroelectric R-stacked bilayer WSe₂" for publication in Nature communications.

Response 2:

We sincerely thank the reviewer for their positive evaluation and their recommendation for publication.

Reviewer #3 (Remarks to the Author):

My concerns in the original manuscript were nicely resolved by the authors in the revised version. Nevertheless, before I can accept the manuscript for publication in Nature Communications, the authors should incorporate the additional data from the reviewers' response into the revised version, at least in the supplementary information. For example, the data in Figures R5 and R6. These will help readers better understand the quality of the sample and the scientific conclusion drawn from the study.

Response 3:

We thank the reviewer for their constructive suggestion. We agree that including these figures improves the transparency and clarity of the study.

As requested, we have incorporated the data from Figures R5 and R6 (from the previous response letter) into the revised Supplementary Information. They can now be found in new **Figure S9** and **Figure S12**, respectively. We believe these additions significantly strengthen the manuscript.

Reply to Reviewers' Comments

Reviewer #1 (Remarks to the Author):

In Response 1 to Reviewer #1's question, the Figure R1 data is considered to be very important. However, this data cannot be found in the revised manuscript or the revised supplementary materials. If Figure R1 and the related discussion are added to the revised manuscript or supplementary, I would recommend the publication of this paper.

Response 1:

We thank the reviewer for the comment. We have now included the data and discussion from Figure R1 of the previous rebuttal into the Supplementary Information as **Figure S3**.

Reviewer #3 (Remarks to the Author):

The authors addressed all of my concerns. I recommend publishing the current version of the manuscript in Nature Communications.

Response 2:

We sincerely thank the reviewer for their recommendation for publication.

Reply to Reviewers' Comments

Reviewer #1 (Remarks to the Author):

I have received sufficient answers to all the questions I asked. I recommend that this paper be published.

Response 1:

We sincerely thank the reviewer for their recommendation for publication.